# The HIF-1α and mTOR Pathways Amplify Heterotopic Ossification

**DOI:** 10.3390/biom14020147

**Published:** 2024-01-24

**Authors:** Haitao Wang, Frederick S. Kaplan, Robert J. Pignolo

**Affiliations:** 1Department of Medicine, Geriatric Medicine & Gerontology, Mayo Clinic, Rochester, MN 55905, USA; 2Department of Physiology and Biomedical Engineering, Mayo Clinic, Rochester, MN 55905, USA; 3Department of Orthopaedic Surgery, The Perelman School of Medicine of the University of Pennsylvania, Philadelphia, PA 19104, USA; 4Department of Medicine, The Perelman School of Medicine of the University of Pennsylvania, Philadelphia, PA 19104, USA; 5The Center for Research in FOP and Related Disorders, The Perelman School of Medicine of the University of Pennsylvania, Philadelphia, PA 19104, USA; 6Department of Medicine, Divisions of Endocrinology, Hospital Internal Medicine, Rochester, MN 55905, USA

**Keywords:** fibrodysplasia ossificans progressiva, hypoxia-inducible factor, mechanistic target of rapamycin (mTOR) pathways, heterotopic ossification

## Abstract

Fibrodysplasia ossificans progressiva (FOP; MIM# 135100) is an ultra-rare congenital disorder caused by gain-of-function point mutations in the Activin receptor A type I (*ACVR1*, also known as *ALK2*) gene. FOP is characterized by episodic heterotopic ossification (HO) in skeletal muscles, tendons, ligaments, or other soft tissues that progressively causes irreversible loss of mobility. FOP mutations cause mild ligand-independent constitutive activation as well as ligand-dependent bone morphogenetic protein (BMP) pathway hypersensitivity of mutant ACVR1. BMP signaling is also a key pathway for mediating acquired HO. However, HO is a highly complex biological process involving multiple interacting signaling pathways. Among them, the hypoxia-inducible factor (HIF) and mechanistic target of rapamycin (mTOR) pathways are intimately involved in both genetic and acquired HO formation. HIF-1α inhibition or mTOR inhibition reduces HO formation in mouse models of FOP or acquired HO in part by de-amplifying the BMP pathway signaling. Here, we review the recent progress on the mechanisms of the HIF-1α and mTOR pathways in the amplification of HO lesions and discuss the future directions and strategies to translate the targeting of HIF-1α and the mTOR pathways into clinical interventions for FOP and other forms of HO.

## 1. Introduction

Heterotopic Ossification (HO) refers to the formation of extraskeletal bone in soft tissues [1,2,3] and represents a significant risk for functional impairment not only to those with causative mutations but also to those with non-monogenetic HO, including civilians who undergo invasive musculoskeletal procedures, those with severe neurological injuries, or those who have experienced massive trauma such as soldiers in modern conflicts.

Classic fibrodysplasia ossificans progressiva (FOP) is an extremely rare congenital disorder caused by a recurrent gain-of-function point mutation (R206H) in the Activin receptor A type I (*ACVR1*, also known as *ALK2*) gene [4]. *ACVR1* is responsible for encoding a receptor involved in the regulation of bone development. In WT ACVR1, the activation of WT ACVR1 needs ligands such as BMP4. The mutation in ACVR1 leads to the receptor becoming sensitive to the Activin A ligand, promoting the formation of bone in soft tissues even in the absence of injury or trauma. Clinically, FOP is characterized by episodic inflammatory exacerbations (called flare-ups) that lead to heterotopic ossification (HO) in skeletal muscles, tendons, or other soft tissues and progressively cause irreversible loss of mobility [2,5].

The nonhereditary or acquired form of HO develops in muscles, tendons, and ligaments. Acquired HO is usually incited upon tissue injury in clinical scenarios such as joint arthroplasty or traumatic brain or spinal cord injuries or as the result of combat-related blasts and burns [6]. Acquired HO can result in persistent symptoms such as pain, swelling, and restricted joint movement when HO affects major joints or muscle groups [7].

Unfortunately, treatment options for genetic and nongenetic HO are limited due to the gaps in knowledge related to the incompletely understood mechanisms of lesion formation. In severe cases, non-steroidal anti-inflammatory drugs (NSAIDs) may be prescribed to manage pain and inflammation. Surgical removal of the ectopic bone may be considered with physical therapy to maintain joint mobility and function. In some instances, HO can recur after surgical removal.

Understanding the underlying molecular mechanisms contributing to the pathogenesis of FOP is crucial for developing effective therapeutic strategies. The Bone Morphogenetic Protein (BMP) pathway is a key pathway for mediating both hereditary and nonhereditary HO formation. In FOP, the canonical FOP mutation ACVR1 (R206H) and all genetic variants reported in humans exhibit loss of the autoinhibition of BMP signaling [4,8]. FOP gain-of-function mutations cause mild constitutive activation and BMP and Activin A-dependent BMP pathway hypersensitivity of ACVR1 [9,10,11,12,13,14].

Importantly, some signaling characteristics are unique to FOP. Activin A, a member of the transforming growth factor-beta (TGF-β)/BMP superfamily, also binds to ACVR1 but normally antagonizes BMP signaling [15,16,17]. In the context of FOP, ACVR1 pathogenic mutations convert Activin A-mediated inhibition into activation of BMP effectors SMAD1/5/8 [15,18]. Consequently, the excessively activated BMP signaling misdirects chondro-osseous differentiation of progenitor cells during tissue repair in damaged sites, leading to the gradual replacement of soft tissues with heterotopic bone [19,20]. Activin A promiscuously stimulates the BMP signaling pathway in vitro and in vivo and mediates HO in FOP mouse models [15,18]. BMP pathway signaling is indispensable for HO formation in FOP. We reported that cotreatment of Activin A and BMP4 synergistically amplify dysregulated BMP signaling and induce chondro-osseous differentiation in Stem cells from human exfoliated deciduous teeth primary cells (SHED) derived from patients with FOP [21].

Recently, the development and approval of palovarotene represents a significant advancement in the search for potential treatments for FOP. Palovarotene is a retinoic acid receptor gamma (RAR-γ) agonist. It works by modulating a specific signaling pathway involved in bone formation and inflammation. By targeting this pathway, it aims to inhibit the aberrant endochondral bone formation seen in FOP. Clinical trial evidence supports palovarotene’s efficacy in reducing new HO in FOP, but there is a high risk of premature physeal closure (PPC) in skeletally immature patients [22]. Additionally, palovarotene does not seem to reduce flare-ups.

This review focuses on the role of two significant pathways, HIF-1α and mTOR, in amplifying HO in FOP and in trauma-induced HO.

## 2. Hypoxia-Inducible Factor 1-Alpha (HIF-1α)

HIF-1α, or Hypoxia-Inducible Factor 1-alpha, is a protein that plays a crucial role in the cellular response to low oxygen levels, a condition known as cellular hypoxia. HIF-1α is a subunit of the HIF-1 complex, which also includes HIF-1β (or ARNT, Aryl hydrocarbon receptor nuclear translocator). In normoxic conditions, HIF-1α is continuously synthesized and degraded. However, when hypoxic conditions exist, such as during high-altitude exposure, inflammation, or in response to injuries, HIF-1α stabilizes and translocates to the cell nucleus [23]. Once in the nucleus, it forms a complex with HIF-1β, and together, they activate the expression of genes that regulate angiogenesis, glycolysis, erythropoiesis, and cell survival [23,24].

HIF-1α promotes angiogenesis, cell survival, and osteogenic differentiation, all of which are critical processes in the formation of heterotopic bone. Inflammatory hypoxia occurs as sites of inflammation undergo significant shifts in metabolic activity leading to oxygen deficiency, caused by the increase in oxygen consumption by the infiltration of immune cells such as monocytes, neutrophils, T cells, and B cells [25,26].

The earliest stages of HO are composed of perivascular inflammatory infiltrates, muscle degeneration, and fibroproliferation [27]. We evaluated the hypoxic response in early FOP lesions from six FOP patients who underwent diagnostic biopsy for presumptive neoplasm prior to the correct diagnosis of FOP. Studies show that HIF-1α was upregulated in lesional tissues from FOP patients, suggesting its possible involvement in FOP pathogenesis [28]. One possible mechanism is that the hypoxic microenvironment created within FOP lesions due to compromised local blood flow from compartment-like syndromes may lead to HIF-1α activation [29]. Cellular hypoxia is one of the primary triggers for flare-ups in FOP. We used control or FOP SHED cells to examine the effect of hypoxia and normoxia on osteogenesis and chondrogenesis. We knocked out HIF-1α expression either genetically or pharmacologically, and we found that cellular hypoxia greatly promoted chondrogenesis, an essential step for endochondral HO. Our investigation supported that cellular oxygen sensing is a critical regulator of HO in FOP [28]. We also reported that the inflammatory molecule lipopolysaccharide (LPS) or Poly I: C induced BMP signaling in SHED cells and Evolutionarily Conserved Signaling Intermediate in the Toll Pathway (ECSIT) linked toll-like receptors (TLR) and BMP pathway signaling [30]. Taken together, our data suggested that inflammation and tissue hypoxia created a microenvironment that amplified and prolonged BMP pathway signaling from mACVR1 and promoted ectopic chondrogenesis and HO in FOP.

Similarly, in trauma-induced HO, the injury-induced stem cell niche is BMP-dependent [31], and the inhibition of HIF-1α also prevented trauma-induced HO [32]. PX-478, a compound that substantially reduces HIF-1α expression, significantly decreased HO formation both in FOP and trauma-induced mouse models [30,32]. Moreover, HIF-1α promoted the expression of vascular endothelial growth factor (VEGF), a potent angiogenic factor, which contributes to the neovascularization observed in FOP lesions [30]. Collectively, these findings suggest that HIF-1α is a key player in the pathological bone formation seen in FOP and nongenetic HO [32].

## 3. HIF-1α Signaling Pathway and BMP Signaling Pathway

While the BMP signaling pathway and the HIF-1α pathway primarily operate through distinct mechanisms, there is evidence of crosstalk and interaction between these pathways in certain cellular contexts. Studies have shown that BMP signaling can modulate the activity of HIF-1α and vice versa. For example, BMP signaling has been implicated in the regulation of HIF-1α stability and activity [33]. BMP signaling can induce the expression of HIF-1α and enhance its stability, leading to increased HIF-1α dependent gene expression under normoxic conditions [34]. These interactions suggest that BMP signaling can influence the cellular response to hypoxia through HIF regulation. Additionally, studies have demonstrated that HIF-1α can affect BMP signaling. HIF-1α increases the intensity and duration of canonical BMP signaling through Rabaptin 5 (RABEP1)-mediated retention of ACVR1 in the endosomal compartment in FOP cells [28]. We further showed that early inflammatory FOP lesions in humans and in a mouse model are markedly hypoxic, and the inhibition of HIF-1α by genetic or pharmacologic means restores canonical BMP signaling to normoxic levels in human FOP cells and profoundly reduces HO in a constitutively active *CA-AVCR1* ^(*Q207D)*^ mouse model of FOP. Upregulation of HIF-1α influences BMP-mediated gene expression, and knockdown of HIF-1α has been shown to inhibit Runx 2, a downstream target of BMP signaling, suggesting that the HIF-1α pathway can impact BMP-dependent processes [35].

In conclusion, BMP signaling and the HIF-1α pathway are both implicated in the pathogenesis of FOP, although the precise molecular mechanisms and crosstalk between these pathways in FOP are still being investigated. Understanding these pathways’ roles in FOP may provide insights into potential therapeutic targets for managing or preventing ectopic bone formation in this debilitating condition and perhaps in other forms of HO.

## 4. mTOR Signaling Pathway in HO

mTOR is a highly conserved serine/threonine kinase that regulates various cellular processes, including cell growth, proliferation, and differentiation [36], via translational regulation of eukaryotic initiation factor 4E (eIF4E) binding protein 1 (4E-BP1) and ribosomal protein S6 kinase 1 (S6K1). mTOR forms two distinct complexes: mTOR Complex 1 (mTORC1) and mTOR Complex 2 (mTORC2) [37]. These complexes have distinct components and functions but are both regulated by cellular signals such as growth factors, energy status, and nutrient availability. Rapamycin and rapamycin-like analogs (rapalogs) are direct inhibitors of the mTOR complex 1 (mTORC1). mTORC1 is sensitive to rapamycin as it contains the regulatory-associated protein of mTOR (Raptor), which is the targeted component [38]. However, mTORC2 is resistant to rapamycin.

Hino et al. identified the role of the mTOR signaling pathway in HO by using a high-throughput screening (HTS) system. mTOR signaling is a pivotal pathway in enhanced chondrogenesis that is initiated by Activin-A and serves as a critical pathway for the aberrant chondrogenesis of mesenchymal stromal cells derived from FOP patient-derived induced pluripotent stem cells (FOP-iPSCs) [39]. In addition, Hino et al. identified ENPP2, an enzyme that generates lysophosphatidic acid, as a linker of FOP-ACVR1 and mTOR signaling in chondrogenesis in FOP-iPSCs with *ACVR1*^R206H^ mutation and wild type. These results uncovered a crucial role of the Activin-A/FOP-ACVR1/ENPP2/mTOR axis in FOP pathogenesis [40].

When inducing HO in the *CA-AVCR1* ^(*Q207D)*^ mouse model, myofiber injury and fibrosis are the initial steps of HO formation. Rapamycin treatment reduces the fibrosis and accumulation of mesenchymal cells observed at the injury site in this mouse *CA-AVCR1 ^(Q207D)^* model [41].

## 5. mTOR and BMP Signaling Pathway

BMP signaling modulates mTOR activity. Studies have suggested that BMP2 induces the phosphorylation and activation of mTOR in lung cancer cells [42]. BMP-2 activates mTORC1 via the mediation of c-Abl, Perk, and Atf4 in the differentiation of bone marrow stromal cells (BMSCs) [43]. BMP signaling is proposed to regulate energy metabolism via mTOR and HIF-1α. mTORC1 plays important roles in regulating energy metabolism and promoting a shift from tricarboxylic acid cycle to glycolysis via HIF-1α [44].

mTOR has also been implicated in the regulation of BMP signaling. mTORC1 influences BMP signaling by regulating the translation of BMP pathway components. For example, rapamycin preserves SMAD 1/5 signaling and osteogenic differentiation in fibroblast cells isolated from the FOP mouse model *CA-AVCR1* ^(*Q207D)*^ [41]. Additionally, Activin A enhances mTOR signaling in FOP [40]. Rapamycin, a direct inhibitor of mTORC1, effectively blocks trauma-induced and genetic HO [32]. Furthermore, mTOR inhibition by rapamycin potently blocks Activin A-induced chondrogenesis in vitro and HO formation in genetically correct FOP model mice [40].

These interactions between BMP signaling and mTOR highlight the intricate cross-regulation between these signaling pathways in cellular processes in HO formation. The report that mTOR inhibition reduces fibrosis and HO through a SMAD 1/5–independent mechanism indicates that increased BMP signaling alone is not responsible for muscle damage in the setting of injury but that concomitant activation of the mTOR pathway leads to fibrosis and ectopic bone formation [41]. mTOR inhibition and BMP signaling can synergistically reduce muscle fibrosis and improve myofiber regeneration [41].

However, it is important to note that the precise mechanisms and consequences of the interplay between the BMP and the mTOR pathways are still an active area of research, and further studies are needed to fully understand their complex interactions.

## 6. Interaction between HIF1-α and mTOR

Emerging evidence suggests crosstalk between the HIF-1α and mTOR pathways, providing a potential mechanism for the amplification of HO in FOP. HIF-1α activates mTOR signaling through various mechanisms, including the upregulation of growth factors and the modulation of metabolic pathways. In addition, hypoxia exposure increases BMP2 expression in cultured osteoblastic cells in an HIF-dependent manner involving activation of the ILK/Akt/mTOR pathway [45].

Conversely, mTOR signaling can stabilize and enhance the transcriptional activity of HIF-1α [46]. The mTOR inhibitor (rapamycin) inhibits the potentiating action of hypoxia [45]. The constitutively active mutant of mTOR induces HIF-1α via the direct substrate of STAT3 in HEK293 cells [47]. The activation of mTOR by Ras homolog enhances the activity of HIF-1α and VEGF during hypoxia [48]. This reciprocal interaction between HIF-1α and mTOR creates a positive feedback loop, leading to sustained activation of both pathways, which further promotes osteogenesis and exacerbates the progression of FOP (Figure 1).

## 7. Treatment Strategies Based on Targeting HIF-1α

Several pharmacological agents, such as HIF-1α inhibitors, have shown promise in pre-clinical studies, highlighting their potential as therapeutic options.

We showed that early inflammatory FOP lesions in humans and in a *CA-AVCR1 ^(Q207D)^* FOP mouse model are markedly hypoxic, and inhibition of HIF-1α by genetic or pharmacologic means restores canonical BMP signaling to normoxic levels in human FOP cells and profoundly reduces HO in a constitutively active *CA-AVCR1* ^(*Q207D)*^ mouse model of FOP. We found that imatinib, a tyrosine kinase inhibitor, decreases HIF-1α activity and mutant ACVR1 activity in hypoxic FOP stem cells. Further, in in vivo experiments, imatinib reduces heterotopic ossification in a mouse model of FOP [28] and a murine post-traumatic model of HO [41]. Importantly, imatinib has been reported to reduce the symptomatology and duration of flare-ups in six out of seven children with unrelenting FOP flare-ups. Imatinib was well-tolerated, with a ubiquitously reported decrease in the intensity of flare-ups in the six children who took the medication [49] (Figure 2).

Using an off-on-off-on (O4) approach to rapidly assess the potential symptomatic efficacy and tolerability of imatinib, three children with classic FOP who had recalcitrant flare-ups were treated with an O4 regimen. Fewer flare-ups, reduced swelling, and improved physical functioning were confirmed by the parents and treating physician when imatinib was given (“on”) compared to when it was held (“off”). A validated tool to prospectively measure and quantitate patient-reported flare-up symptoms is presently being developed and validated. Similar to the previous case reports, imatinib improved flare-up symptomatology with a median of 2–3 weeks [50]. This O4 approach did not claim, nor is there any evidence to support or refute, that imatinib has any effect on HO in FOP. The clinical findings described in this report support the design of a brief and adaptive placebo-controlled trial to assess the potential efficacy of imatinib in reducing the symptoms in children with refractory flare-ups of FOP [50].

In a recent study, siRNA knockdowns of HIF-1α mRNA and -Runx2 mRNA resulted in the inhibition of HO in an Achilles tenotomy mouse model [35]. The inhibitory effect may be attributed to the inhibition of angiogenesis concomitantly with bone and cartilage-related gene expression due to HIF-1α siRNA [35]. Notably, there is a synergistic inhibitory effect of HO formation when silencing of both HIF-1α and Runx2 is achieved [35].

## 8. Treatment Strategies Based on mTOR as a Target

After screening 500 small molecule compounds using the HTS screening system on a chondrogenic cell line with FOP-ACVR1 expression [40], Hino et al. identified that TAK 165 (also known as mubritinib) inhibits glycosaminoglycan (GAG) production, a critical component of the extracellular matrix required for the requisite chondrogenic stage of HO lesion formation [39]. TAK 165 can upregulate intracellular ROS concentration and cell apoptosis, reduce the activation of the PI3K/mTOR signaling pathway, and disrupt mitochondrial function [51]. In this study, the mitochondrial respiratory chain complex enzyme I, III, IV, and V involved in ATP energy synthesis showed a significant decrease after TAK 165 treatment (*p*  <  0.05), suggesting that mubritinib can cause mitochondrial dysfunction and lead to abnormal energy metabolism [51]. Moreover, TAK 165 effectively decreases mineralization and Collagen I alpha gene expression and suppresses HO in an FOP mouse model and in a BMP-7-induced HO model using WT mice [39]. There appear to be no observable side effects of TAK 165 dosing in mice, and they maintain normal body weight, normal chondrogenesis, and normal skeletal development in vivo. In addition, rapamycin (RAPA), as mentioned above, reduces the vascularization triggered by mTOR signaling and inhibits HO formation [51]. Inhibition of these required pathways may likely contribute to abrogation of HO formation. Of note, rapamycin causes muscle atrophy in models of muscle trauma [52]. However, in the *CA-AVCR1* ^(*Q207D*^*^)^* mouse model with hyperactive BMP signaling, myofiber size improved or preserved even with rapamycin treatment [41].

## 9. Future Directions

Inflammation and the cellular oxygen-sensing mechanism modulates intracellular retention of a mutant BMP receptor and determines, in part, its pathologic activity in FOP. This analysis provides critical insight into the role of HIF-1α in the hypoxic amplification of BMP signaling and in the episodic induction of HO in FOP and further identifies HIF-1α as a therapeutic target for FOP and perhaps nongenetic forms of HO [28]. Importantly, the intricate interplay between the HIF-1α and mTOR pathways appears to play a crucial role in amplifying HO. The upregulation and interaction of these pathways likely contributes to the dysregulated bone formation observed in FOP patients and in those with nonhereditary HO. Therefore, targeting HIF-1α and mTOR signaling pathways may hold promising therapeutic potential in mitigating HO progression and improving the quality of life for patients with HO due to multiple etiologies. Further research is necessary to unravel the exact mechanisms governing this intricate link between hypoxia and inflammation and the mTOR pathway involved in early HO lesion formation. Additionally, combination therapies targeting both pathways may provide synergistic effects in suppressing HO. In the case of FOP, targeting the ALK2 kinase and associated HIF-1α and mTOR signaling pathways may have additive or even synergistic benefits.

## Figures and Tables

**Figure 1 biomolecules-14-00147-f001:**
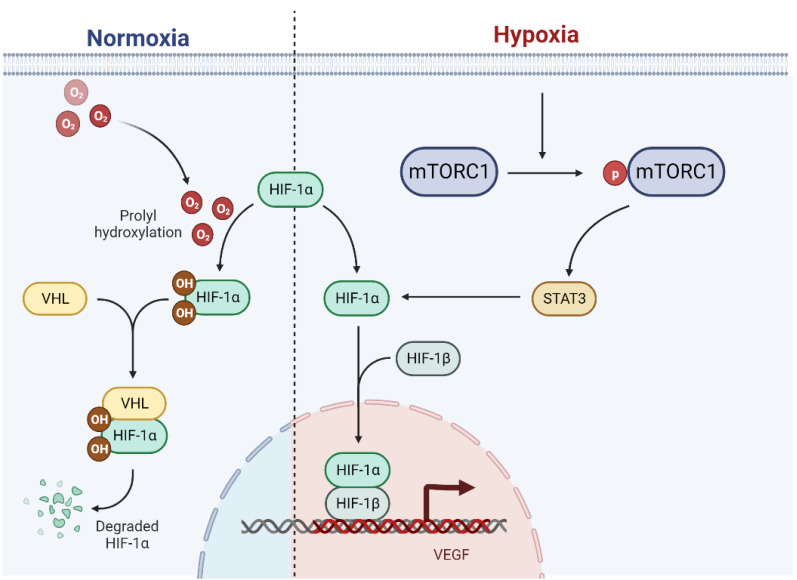
Interactions between the HIF pathway and mTOR pathway under hypoxia. Under normoxia, HIF-1α is expressed constitutively and hydroxylated by prolyl hydroxylases. Then, HIF-1α is recognized by the von Hippel–Lindau tumor-suppressor protein (VHL) and degraded. Under hypoxia, the catalytic activity of prolyl hydroxylases is inhibited. HIF-1α accumulates in the nucleus and dimerizes with HIF-β, and together, they promote the transcription of hypoxia-response genes. Hypoxia also induces the phosphorylation of MTORC1, and the activation of the mTOR pathway can reciprocally stabilize HIF-1α via STAT3. The activation of HIF-1α and VEGF will further promote HO formation.

**Figure 2 biomolecules-14-00147-f002:**
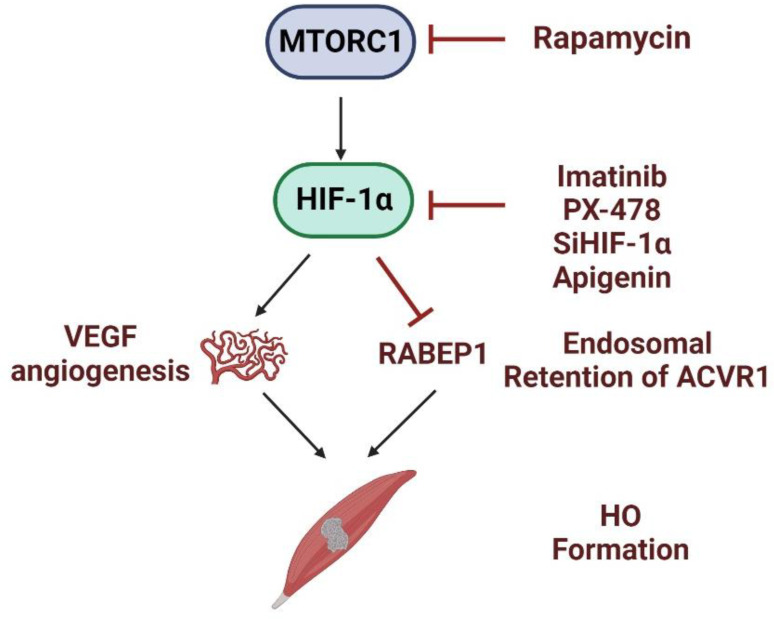
Inhibitor of mTOR and HIF pathways. Rapamycin inhibits the mTOR pathway. Imatinib, PX-478, apigenin, and HIF1-α siRNA inhibit HIF1-α. Together, inhibition of these pathways blocks angiogenesis and increases the retention of the ACVR1 gene in endosomal compartments and abrogates HO formation.

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
