# Peer review of "The HIF-1α and mTOR Pathways Amplify Heterotopic Ossification"

_biomolecules, 2024, doi:10.3390/biom14020147_

Round 1

Reviewer 1 Report

Comments and Suggestions for Authors

The review explores the involvement of HIF and mTOR pathways in the initiation and progression of n acquired and genetic heterotopic ossification i, highlighting their interaction with the BMP pathways. However, the organization of the manuscript leaves room for improvement, as it lacks coherence and readability. There is a recurrence of concepts, and several crucial aspects are overlooked. 

Introduction:

An immediate reference to palovarotene as a potential therapeutic for FOP appears abruptly in the introduction (line 43). To enhance clarity, it is suggested to relocate this reference after detailing the mutation and main signaling. Since the focus is not on RAR pathways in the review, presenting this strategy among the approved ones (with a note on activin inhibitors, which are not mentioned) would provide a more comprehensive overview. This underscores the need to better characterize other targets for improved treatment options.

(Line 79-82)

The emphasis on a specific study at this point is unnecessary, considering the numerous studies exploring Activin-A-BMP in FOP.

Chapters on HIF:

The manuscript lacks information on the cells contributing to hypoxia and those in which the HIF pathway is active. Furthermore, there is no mention of the role of the immune system or vasculogenesis throughout the manuscript. It is recommended to combine the paragraphs on HIF and BMP, followed by a discussion of treatment strategies.

Chapter 4: The Mechanistic Target of Rapamycin (mTOR):

The title should be revised to "mTOR Signaling Pathway in Heterotopic Ossification (HO)," emphasizing the distinction between rapamycin and mTOR. The discussion on the crosstalk with the BMP2 pathway should be integrated into the same paragraph, followed by an exploration of treatment strategies involving rapamycin and emerging drugs. Additionally, the manuscript lacks mention of the specific cell niche where these events occur, despite the substantial impact of rapamycin on various cell types, including wild-type and R206H mutant cells.

The final paragraph should focus on the crosstalk between HIF and mTOR, leading to future perspectives.

Overall, restructuring and enhancing the content in these areas will contribute to a more cohesive and comprehensible manuscript.

Author Response

Dear Reviewer,

Our gratitude extends to the reviewers for the invaluable comments, which have significantly strengthened our manuscript. We are pleased to present a revised version of our manuscript for your consideration. Here is the answer we highlighted in yellow.

Thanks.

The review explores the involvement of HIF and mTOR pathways in the initiation and progression of n acquired and genetic heterotopic ossification i, highlighting their interaction with the BMP pathways. However, the organization of the manuscript leaves room for improvement, as it lacks coherence and readability. There is a recurrence of concepts, and several crucial aspects are overlooked. 

Introduction:

An immediate reference to palovarotene as a potential therapeutic for FOP appears abruptly in the introduction (line 43). To enhance clarity, it is suggested to relocate this reference after detailing the mutation and main signaling. Since the focus is not on RAR pathways in the review, presenting this strategy among the approved ones (with a note on activin inhibitors, which are not mentioned) would provide a more comprehensive overview. This underscores the need to better characterize other targets for improved treatment options.

Thanks. We moved the discussion of palovarotene to the end of the Introduction.

 (Line 79-82)

The emphasis on a specific study at this point is unnecessary, considering the numerous studies exploring Activin-A-BMP in FOP.

Thanks. Activin A or BMPs can stimulate HO; However, in FOP, treatment of Activin A and BMP4 can synergistically boost BMP signaling in mesenchymal progenitor cells.  We added the co-treatment to indicate the significance.

Sections  on HIF:

The manuscript lacks information on the cells contributing to hypoxia and those in which the HIF pathway is active. Furthermore, there is no mention of the role of the immune system or vasculogenesis throughout the manuscript. It is recommended to combine the paragraphs on HIF and BMP, followed by a discussion of treatment strategies.

We added the discussion of inflammation hypoxia, immune cells, and vasculogenesis to the HIF chapter  since we already had a combined paragraph of HIF and BMP. To maintain the structure of the review and emphasize the crosstalk, we kept the crosstalk paragraph independently. 

Chapter 4: The Mechanistic Target of Rapamycin (mTOR):

The title of this section should be revised to "mTOR Signaling Pathway in Heterotopic Ossification (HO)," emphasizing the distinction between rapamycin and mTOR. The discussion on the crosstalk with the BMP2 pathway should be integrated into the same paragraph, followed by an exploration of treatment strategies involving rapamycin and emerging drugs. Additionally, the manuscript lacks mention of the specific cell niche where these events occur, despite the substantial impact of rapamycin on various cell types, including wild-type and R206H mutant cells.

We changed the title of this section to “mTOR Signaling Pathway in HO”. However, to maintain the structure of the review and emphasize the cross-talk between the HIF and mTOR pathways,  we kept the crosstalk paragraph independently.

FOP-iPSCs with wild-type and R206H mutant are added.

The final paragraph should focus on the crosstalk between HIF and mTOR, leading to future perspectives.

We focused on the crosstalk of HIF and mTOR.

 Overall, restructuring and enhancing the content in these areas will contribute to a more cohesive and comprehensible manuscript.

Thanks for your comments. We revised it accordingly. 

Reviewer 2 Report

Comments and Suggestions for Authors

Abstract

·         The abstract is well-written, providing a clear overview of the article's content. It effectively communicates the focus on the HIF-1α and mTOR pathways in amplifying heterotopic ossification (HO).

1.      Introduction

·         In Line 29: “Heterotopic Ossification (HO) refers to the formation of extraskeletal bone in soft tissues (1).” The author cite a paper from 2019. I would recommend citing the original or earliest paper that describes HO. Several papers published in the 80s describe HO.

·         In line 31: Replace “nongenetic HO” with non-monogenetic HO.

·         In line 52-53: The sentence appears to be incomplete, “The nonhereditary or acquired form, in which HO develops in muscle, tendon, and ligament”. I would recommend rephrasing.

·         Line 55-62: Please provide references.

2.      Hypoxia-inducible factor 1-alpha (HIF-1α )

·         Lines 95-103: Provide some details on the studies mentioned, especially those related to the upregulation of HIF-1α in FOP lesions. Elaborate on the methods and significance of these findings to enhance the reader's understanding.

3.      HIF-1α signaling pathway and BMP signaling pathway

·         The passage is well-organised, with a clear structure that guides the reader through the interactions between BMP signaling and the HIF-1α pathway in FOP. The logical flow contributes to easy comprehension.

·         Line 127: Review font and size of “CA-AVCR1”

4.      The mechanistic target of rapamycin (mTOR)

·         Consider providing brief explanations for less common abbreviations, such as FOP-iPSCs and FOP-iMSCs, to enhance accessibility.

5.      mTOR and BMP signaling pathway

·         In line 168: Authors state that rapamycin activates Smads 1 and 5 in human prostate cancer cells and tissues by inhibiting mTORC1 kinase3. However, how this is relevant to FOP is unclear. The authors assert this observation without establishing its significance or connection to FOP pathogenesis. Further clarification is needed to bridge this gap in understanding.

6.      Interaction between HIF1-α and mTOR

·         The information is well-organized, with a clear presentation of the reciprocal interaction between HIF-1α and mTOR. The figure contributes to the clarity of the content.

·         Line 201: review font and size of “Then”.

7. Treatment strategies based on targeting HIF-1α

·         Line 211: Please provide more information on “genetically-correct FOP mouse model”. What mice model was used?

·         Line 227: The passage introduces the O4 approach for assessing imatinib's efficacy in FOP, but it lacks information on whether the researchers addressed potential biases introduced by self-reporting. Addressing this concern by providing details on methodological rigor, measures to mitigate bias, and transparency in reporting would strengthen the credibility of the study findings.

8. Treatment strategies based on mTOR as a target

·         Line 243: The statement "TAK 165 is an mTOR signaling modulator that indirectly controls mTOR signaling" is somewhat ambiguous and might benefit from more specific information about the mechanism involved. Providing more context on how TAK 165 influences mTOR signaling and connecting it with the mentioned reduction in PI3K/mTOR signaling and disruption of mitochondrial function would improve the overall clarity and coherence of the passage.

9. Future Directions

·         The information is presented in a clear and structured manner, highlighting the significance of HIF-1α as a therapeutic target for FOP and potentially non-genetic forms of HO. It also emphasises the crucial role of the interplay between the HIF-1α and mTOR pathways.

·         Line 257: It would be beneficial to include specific publication details or sources for the study mentioned as "Our study".

Comments on the Quality of English Language

N/A

Author Response

Dear Reviewer,

Our gratitude extends to the reviewers for the invaluable comments, which have significantly strengthened our manuscript. We are pleased to present a revised version of our manuscript for your consideration. Here is the answer we highlighted in yellow.

Thanks.

Abstract

  • The abstract is well-written, providing a clear overview of the article's content. It effectively communicates the focus on the HIF-1α and mTOR pathways in amplifying heterotopic ossification (HO).
  1. Introduction
  • In Line 29: “Heterotopic Ossification (HO) refers to the formation of extraskeletal bone in soft tissues (1).” The author cite a paper from 2019. I would recommend citing the original or earliest paper that describes HO. Several papers published in the 80s describe HO.

Added

  • In line 31: Replace “nongenetic HO” with non-monogenetic HO.

Replaced

  • In line 52-53: The sentence appears to be incomplete, “The nonhereditary or acquired form, in which HO develops in muscle, tendon, and ligament”. I would recommend rephrasing.

Changed.

  • Line 55-62: Please provide references.

Added

  1. Hypoxia-inducible factor 1-alpha (HIF-1α )
  • Lines 95-103: Provide some details on the studies mentioned, especially those related to the upregulation of HIF-1α in FOP lesions. Elaborate on the methods and significance of these findings to enhance the reader's understanding.

Added

  1. HIF-1α signaling pathway and BMP signaling pathway
  • The passage is well-organised, with a clear structure that guides the reader through the interactions between BMP signaling and the HIF-1α pathway in FOP. The logical flow contributes to easy comprehension.
  • Line 127: Review font and size of “CA-AVCR1”

Changed

  1. The mechanistic target of rapamycin (mTOR)
  • Consider providing brief explanations for less common abbreviations, such as FOP-iPSCs and FOP-iMSCs, to enhance accessibility.

Added

  1. mTOR and BMP signaling pathway
  • In line 168: Authors state that rapamycin activates Smads 1 and 5 in human prostate cancer cells and tissues by inhibiting mTORC1 kinase3. However, how this is relevant to FOP is unclear. The authors assert this observation without establishing its significance or connection to FOP pathogenesis. Further clarification is needed to bridge this gap in understanding.

We removed this sentence; the next sentence is more relevant to FOP and delivers a similar message.

  1. Interaction between HIF1-α and mTOR
  • The information is well-organized, with a clear presentation of the reciprocal interaction between HIF-1α and mTOR. The figure contributes to the clarity of the content.
  • Line 201: review font and size of “Then”.

Changed.

  1. Treatment strategies based on targeting HIF-1α
  • Line 211: Please provide more information on “genetically-correct FOP mouse model”. What mice model was used?

We provided the mouse model instead.

  • Line 227: The passage introduces the O4 approach for assessing imatinib's efficacy in FOP, but it lacks information on whether the researchers addressed potential biases introduced by self-reporting. Addressing this concern by providing details on methodological rigor, measures to mitigate bias, and transparency in reporting would strengthen the credibility of the study findings.

We added the limitations to the discussion as well as  the validation tool of parent reports to be prospectively measured by physicians.

  1. Treatment strategies based on mTOR as a target
  • Line 243: The statement "TAK 165 is an mTOR signaling modulator that indirectly controls mTOR signaling" is somewhat ambiguous and might benefit from more specific information about the mechanism involved. Providing more context on how TAK 165 influences mTOR signaling and connecting it with the mentioned reduction in PI3K/mTOR signaling and disruption of mitochondrial function would improve the overall clarity and coherence of the passage.

Added

  1. Future Directions
  • The information is presented in a clear and structured manner, highlighting the significance of HIF-1α as a therapeutic target for FOP and potentially non-genetic forms of HO. It also emphasises the crucial role of the interplay between the HIF-1α and mTOR pathways.
  • Line 257: It would be beneficial to include specific publication details or sources for the study mentioned as "Our study".

The publication is added to the end of the sentence.